# The epidemiology and outcomes of central nervous system infections in Far North Queensland, tropical Australia; 2000-2019

**Hannah Gora**[1]*, **Simon Smith**[2], **Ian Wilson**[2], **Annie Preston-Thomas**[3], **Nicole Ramsamy**[4], **Josh Hanson**[2,5]

**1** College of Medicine and Dentistry, James Cook University, Cairns, Queensland, Australia, **2** Department of Medicine, Cairns Hospital, Cairns, Queensland, Australia, **3** Tropical Public Health Services, Cairns, Queensland, Australia, **4** Weipa Integrated Health Service, Weipa, Queensland, Australia, **5** The Kirby Institute, University of New South Wales, Kensington, New South Wales, Australia

* hannahaltaf.gora@my.jcu.edu.au

**Data Availability Statement:** All relevant data are within the manuscript and its Supporting Information files.

## Abstract

### Background

The epidemiology of central nervous system (CNS) infections in tropical Australia is incompletely defined.

### Methods

A retrospective study of all individuals in Far North Queensland, tropical Australia, who were diagnosed with a CNS infection between January 1, 2000, and December 31, 2019. The microbiological aetiology of the infection was correlated with patients' demographic characteristics and their clinical course.

### Results

There were 725 cases of CNS infection during the study period, meningitis (77.4%) was the most common, followed by brain abscess (11.6%), encephalitis (9.9%) and spinal infection (1.1%). Infants (24.3%, p<0.0001) and Aboriginal and Torres Strait Islander Australians (175/666 local residents, 26.3%, p<0.0001) were over-represented in the cohort.

A pathogen was identified in 513 cases (70.8%); this was viral in 299 (41.2%), bacterial in 175 (24.1%) and fungal in 35 (4.8%). Cryptococcal meningitis (24 cases) was diagnosed as frequently as pneumococcal meningitis (24 cases). There were only 2 CNS infections with a *S. pneumoniae* serotype in the 13-valent pneumococcal vaccine after its addition to the National Immunisation schedule in 2011. Tropical pathogens–including *Cryptococcus species* (9/84, 11%), *Mycobacterium tuberculosis* (7/84, 8%) and *Burkholderia pseudomallei* (5/84, 6%)–were among the most common causes of brain abscess. However, arboviral CNS infections were rare, with only one locally acquired case—a dengue infection in 2009—diagnosed in the entire study period. Intensive Care Unit admission was necessary in 14.3%; the overall case fatality rate was 4.4%.

**Funding:** This research received no specific funding.

**Competing interests:** The authors have declared that no competing interests exist.

## Conclusion

Tropical pathogens cause CNS infections as commonly as traditional bacterial pathogens in this region of tropical Australia. However, despite being highlighted in the national consensus guidelines, arboviruses were identified very rarely. Prompt access to sophisticated diagnostic and supportive care in Australia's well-resourced public health system is likely to have contributed to the cohort's low case-fatality rate.

## Introduction

Infections of the central nervous system (CNS) result in significant global morbidity and mortality, however, early diagnosis and prompt, targeted treatment can improve outcomes. While a definitive diagnosis is awaited, it is essential to ensure that optimal empirical antimicrobial therapy is administered. However, this is only possible if clinicians have a good understanding of the likely local pathogens [1].

The epidemiology of CNS infections in Far North Queensland (FNQ), in tropical Australia, is influenced by environmental, pathogen and human factors. The region's tropical climate, vectors (including ticks, mites, and mosquitoes), amplifying intermediate hosts, significant agricultural industry and surrounding harsh environment might all be expected to contribute to the local burden of disease [2–5]. FNQ shares a maritime border with Papua New Guinea (PNG) and since the establishment of the Torres Strait treaty in 1985, residents of PNG and the outer Torres Strait Islands have been able to move freely across the border to maintain cultural ties. However, there is a higher prevalence of many infectious diseases—including tuberculosis, dengue, malaria, and Japanese encephalitis virus (JEV)—in PNG, which may be imported into Australia [6]. Although these pathogens have not become established on the Australian mainland, imported cases have been reported [7,8]. These infections have the potential to cause life-threatening disease in some of the most remote communities in Australia.

Urban expansion from recent population growth in the region (with a resulting expansion in medical services leading to more immunocompromised patients) and the area's reputation as an international travel hub, might also be expected to affect the incidence of disease [2,3,9,10]. Approximately 17% of the local population identifies as an Aboriginal and/or Torres Strait Islander Australian [11]. Many members of these Indigenous communities experience significant socioeconomic disadvantage and, accordingly, a high burden of comorbidities, increasing their susceptibility to infection, including CNS disease [12–15].

Conversely, several recent public health interventions might be expected to lessen the local burden of CNS infections. A significant recent expansion of Australia's national vaccination programme would be anticipated to reduce the local incidence of *Streptococcus pneumoniae* and *Neisseria meningitidis* infections, building on the progress seen against *Haemophilus influenzae* in the late 20th century [16]. Introduction of the herpes zoster vaccine in 2005 and expanded influenza and measles/mumps/rubella vaccine coverage might also be expected to have a salutary effect [17–19]. The local public health unit's implementation of arbovirus vector control interventions, including surveillance and control of mosquito populations and release of *Wolbachia*-infected mosquitoes–which mitigate the transmission of dengue, zika and chikungunya–would be expected to have a positive impact [20–22]. Meanwhile, the Queensland State Government sponsored Aboriginal and Torres Strait Islander Environmental Health Program, introduced in 2002, aims to target deficiencies in health infrastructure for

the local Aboriginal and Torres Strait Islander population, particularly in remote locations [23]. The program consists of a suite of public health interventions that address housing, water, sanitation, waste disposal and animal management, all of which have the potential to influence the burden of CNS infections [24,25].

This study was performed to define, more precisely, the temporospatial epidemiology of CNS infections in the FNQ region and to identify the association between demographic factors—including age, Indigenous status, and remote residence—and the aetiology and clinical course of the infections. It also intended to characterise the spectrum and burden of imported pathogens from neighbouring PNG. The study aimed to determine the success of recent public health interventions including changes to the national immunisation schedule and local arbovirus vector control measures on the incidence of CNS infections. These data might be used to inform public health strategies to reduce the local burden of potentially life-threatening CNS infections in the future.

## Methods

This retrospective study was performed at Cairns Hospital in Far North Queensland, Australia. Cairns Hospital is a 531-bed tertiary-referral hospital serving a population of approximately 280,000 people who live in an area of over 380,000 km$^2$ (Fig 1) [26]. All patients with a diagnosis of CNS infection between January 1, 2000, and December 31, 2019, were eligible for inclusion in the study. The Australian Society of Infectious Diseases and United States' Center for Disease Control and Prevention guidelines were used for the case definitions of CNS infections (S1 File) [27,28]. Cases were identified by reviewing all hospital admissions during the study period with relevant International Classification of Diseases (ICD) coding. Additional cases were identified by searching Cairns Hospital's electronic laboratory database for any cerebrospinal fluid (CSF) in which an organism was identified by culture, PCR, antigen, or antibody detection and by reviewing the infectious diseases department's database of cases. Repeat samples were not included. *Plasmodium falciparum* is not considered a cause of meningitis or encephalitis, but it may have a similar clinical presentation [29]. As one of the aims of the study was to determine if CNS pathogens were being imported from PNG, a country which has one of the highest incidences of malaria in the region, patients with cerebral malaria (impaired consciousness and laboratory confirmed *P. falciparum* infection) were also identified [30]. As the consensus guidelines for the investigation and management of encephalitis in adults and children in Australia and New Zealand suggest that autoimmune encephalitis is responsible for up to a third of patients presenting with an encephalitis clinical syndrome [27], patients meeting the case definitions of autoimmune encephalitis from a recent consensus statement were also sought (S2 File) [31].

Laboratory data and radiology reports were retrieved from the public health system's pathology database and imaging system. Additional clinical information–including the patients' demographics, clinical features, comorbidities, and outcomes–were collected from the medical records of cases between January 1, 2015, and December 31, 2019. This period was chosen as it coincided with the introduction of an electronic medical record, facilitating data collection. The patients' Indigenous status was also recorded: all individuals receiving care in Queensland's public health system are asked whether they identify as an Aboriginal Australian, a Torres Strait Islander Australian, both, or neither. Infants were defined as those aged <1 year, children as those aged 1 to 17 years and adults as those aged ≥18 years. The Australian Statistical Geographical Classification Remoteness Area (ASGC-RA) framework was used to determine the remoteness of areas in Australia. Urban areas were defined as those within categories ASGC-RA 1 to 3, and remote areas as those within ASGC-RA 4 and 5 [11]. A patient was defined as being immunocompromised if they had confirmed human immunodeficiency virus (HIV) infection

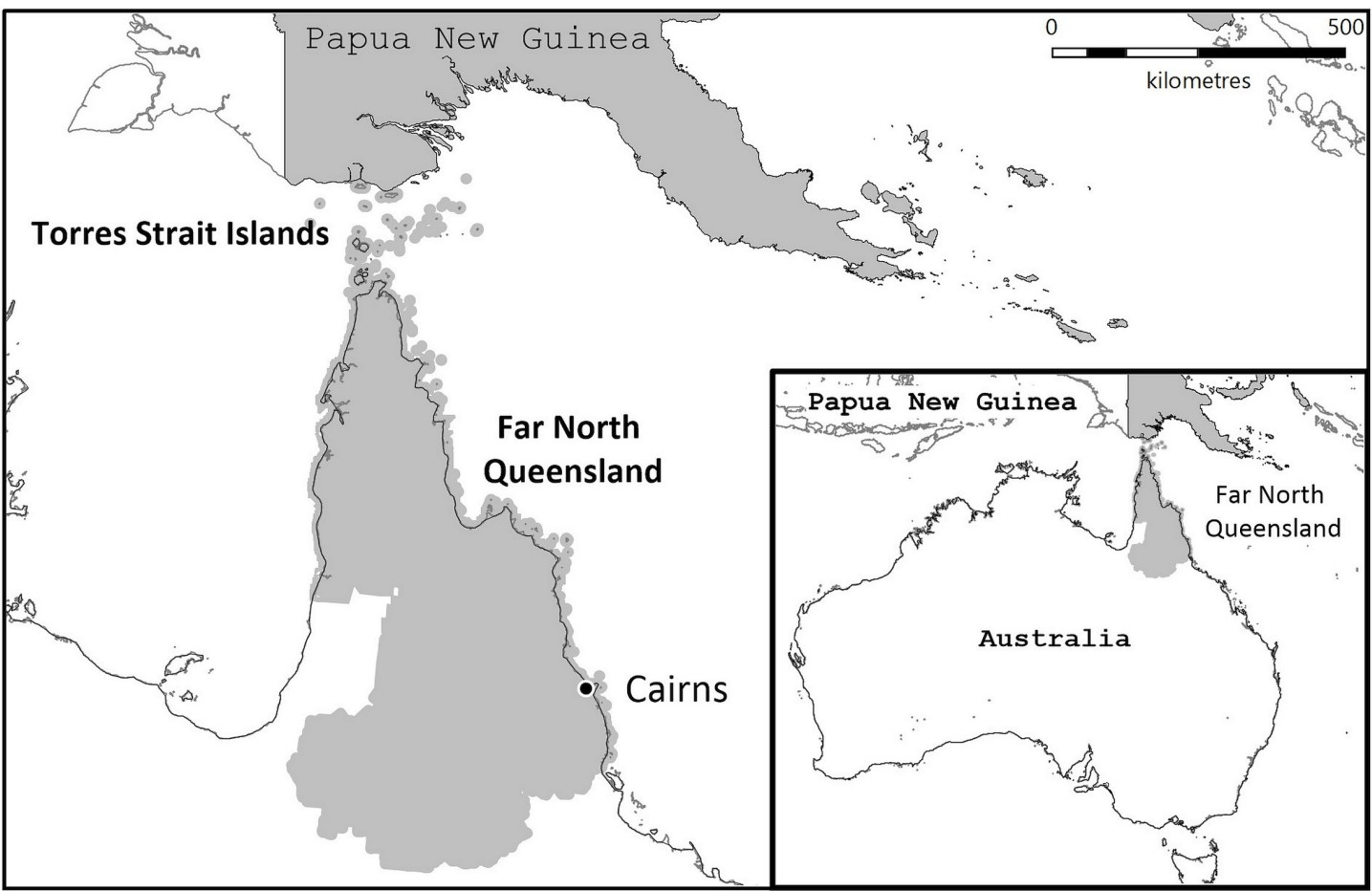

**Fig 1. Map Far North Queensland, Australia, highlighting its proximity to Papua New Guinea.** The map was constructed using mapping software (MapInfo version 15.02, Connecticut, USA) using data provided by the State of Queensland (QSpatial). Queensland Place Names—State of Queensland (Department of Natural Resources, Mines and Energy) 2019, available under Creative Commons Attribution 4.0 International licence https://creativecommons.org/licenses/by/4.0/. 'Coastline and state border–Queensland—State of Queensland (Department of Natural Resources, Mines and Energy) 2019, available under Creative Commons Attribution 4.0 International licence https://creativecommons.org/licenses/by/4.0/.

or were receiving systemic immunosuppressive therapy [32]. An unusual pathogen was defined as one that was not covered by the empirical treatment regime outlined in the Australian Therapeutic Guidelines [33]. Tropical pathogens included pathogens typically seen in tropical climates. Disability was defined as a Modified Rankin Scale score between 2 and 5 –or the requirement for an anticonvulsant medication–on hospital discharge [34]. The management of the patients was said to be appropriate if it was concordant with Australian Therapeutic Guidelines for the identified pathogen, or the clinical syndrome if no pathogen was identified [33]. Australian Bureau of Statistics population data were used to determine disease incidence [26].

## Data analysis

All data were de-identified, entered into an electronic dataset (Microsoft Excel, S1 Dataset) and analysed with statistical software (Stata v 14.2). Univariate analysis was performed using the Kruskal-Wallis, Fisher's exact and chi-squared tests where appropriate. Multivariate analysis was performed using logistic regression. Trends over time were determined, using year as a continuous variable, using an extension of the Wilcoxon rank sum test [35].

### Ethics review

The Far North Queensland Human Research Ethics Committee provided ethical approval for this study (HREC/2020/QCH/59103–1428) and waived the requirement for informed patient consent given the retrospective nature of the study, and de-identified nature of the data.

## Results

Between January 1, 2000, and December 31, 2019, a total of 859 potential cases of CNS infection were identified, however, after review, 134 failed to meet inclusion criteria, leaving 725 cases for analysis. This included 561 (77.4%) cases of meningitis, 84 (11.6%) cases of brain abscess, 72 (9.9%) cases of encephalitis and 8 (1.1%) cases of spinal cord disease (myelitis or intraspinal abscesses) (S1 Fig).

The incidence of CNS infections in the FNQ residents was 13.0/100,000 population in 2000 compared to 21.6/100,000 population in 2019, however the change in incidence did not reach statistical significance (p for trend = 0.09) (S1 Table). Of the 725 cases, 401 (55.3%) occurred in males; 425 (58.6%) occurred in adults and 176 (24.3%) occurred in infants (Table 1). A causative organism was identified in 513/725 (70.8%). Bacterial infection occurred in 175/725 (24.1%) cases, viral infection in 299/725 (41.2%) cases and fungal infection in 35/725 (4.8%) cases. Viral infection was most commonly responsible in infants and adults, although bacteria were more common in children (Table 1).

### Aboriginal and Torres Strait Islander people

Aboriginal and Torres Strait Islander status was available in 702 cases, 178 (25.4%) of whom identified as Indigenous. Among the 666 local FNQ residents, 175 (26.3%) identified as Indigenous Australians compared with 49241/287107 (17.2%) of the general FNQ population at the end of the study period (p<0.0001). Local Indigenous residents with CNS infection were more likely to live in a remote location than local non-Indigenous people (55/175 (31.4%) versus 24/491 (4.9%), p<0.0001). The median (IQR) age of the Indigenous patients was 13 (0–35) years compared with 26 (3–48) years among non-Indigenous patients (p = 0.0001). Indigenous patients with CNS infection were more likely to be infants (p = 0.003). The proportion of patients with the different clinical phenotypes was similar for Indigenous Australians and

**Table 1. The number of cases of CNS infection in infants, children and adults stratified by clinical phenotype and aetiology.**

|  | Infant | Child | Adult | Total |
|---|---|---|---|---|
| **All** | 176 (24.3%) | 124 (17.1%) | 425 (58.6%) | 725 |
| **Clinical phenotypes** |  |  |  |  |
| **Meningitis** | 165 (93.8%) | 100 (80.7%) | 296 (69.7%) | 561 (77.4%) |
| **Encephalitis** | 7 (4.0%) | 8 (6.5%) | 57 (13.4%) | 72 (9.9%) |
| **Brain abscess** | 4 (2.3%) | 15 (12.1%) | 65 (15.3%) | 84 (11.6%) |
| **Spinal disease** | 0 | 1 (0.8%) | 7 (1.7%) | 8 (1.1%) |
| **Aetiological agent** |  |  |  |  |
| **Bacterial** | 36 (20.4%) | 44 (35.5%) | 95 (22.4%) | 175 (24.1%) |
| **Viral** | 86 (48.9%) | 43 (34.7%) | 170 (40.0%) | 299 (41.2%) |
| **Fungal** | 0 | 1 (0.8%) | 34 (8.0%) | 35 (4.8%) |
| **Other** [a] | 0 | 0 | 4 (0.9%) | 4 (0.6%) |
| **Not identified** | 54 (30.7%) | 36 (29.0%) | 122 (28.7%) | 212 (29.2%) |

[a] Two protozoan, one parasitic and one amoebic.

non-Indigenous Australians, but viral aetiologies were diagnosed less commonly—and fungal aetiologies were diagnosed more commonly—in Indigenous Australians (Table 2). Cryptococcal and meningococcal infection were more common in Indigenous Australians than in non-Indigenous Australians. The higher rate of pneumococcal CNS infection in Indigenous Australians did not reach statistical significance. The rates of vaccine-preventable disease–including *S. pneumoniae*, *N. meningitidis*, and *H. influenzae* infection–were higher in Indigenous children and infants (p<0.0001) (S2 Table). There was only a single case of CNS tuberculosis in an Indigenous Australian during the entire study period (Table 3). The 5 additional cases of CNS tuberculosis in FNQ residents occurred in a non-Indigenous Australian, 2 patients born in PNG, 1 patient born in Laos and 1 patient born in England.

## Impact of remote residence on disease incidence

A residential address was available in 679 local FNQ residents: there were 80 (11.8%) patients from remote areas; compared to 29953/286995 (10.4%) of the general FNQ population at the end of the study period (p = 0.25). *B. pseudomallei* was the only pathogen that was more likely to occur in a resident living a remote location (Table 4). A total of 5356 lumbar punctures were performed in FNQ during the study period: 4562 (85.2%) were performed in urban areas; 153951/279459 (55.1%) of the general FNQ population lived in an urban area in the 2016 Australian census (p<0.0001).

The cohort contained 32 PNG residents; *Mycobacterium tuberculosis* was the most common isolate in these individuals (7/32, 21%), compared to only a single case of CNS tuberculosis in an Australian resident (1/679, 0.1%) (p<0.001). There were 767 imported cases of *P. falciparum* malaria diagnosed in FNQ during the study period, of which only 83/767 (10.8%) occurred after 2010. In the last five years of the study period, the period where clinical data were available, only 1/29 *P. falciparum* cases, a PNG resident, had impaired consciousness. Among these 29 *P. falciparum* cases, 12 (41%) were from PNG, 11 (38%) from Africa, 4 (14%) from Indonesia, 1 (3%) from the Solomon Islands and 1 (3%) was from India. Of the 29 cases, 18 (62%) were returning travelers, 8 (28%) were refugees or migrants, and 3 (10%) were PNG nationals transferred to Cairns for further management.

**Table 2. The number of cases in Indigenous and non-Indigenous Australians stratified by age, clinical phenotypes and aetiological agent.**

| | All n = 702[a] | Indigenous n = 178 | Non-Indigenous n = 524 | p |
|---|---|---|---|---|
| **Age group** | | | | |
| **Infants** | 167 (23.8%) | 57 (32.0%) | 110 (21.0%) | 0.003 |
| **Children** | 117 (16.7%) | 41 (23.0%) | 76 (14.5%) | 0.008 |
| **Adults** | 418 (59.5%) | 80 (44.9%) | 338 (64.5%) | <0.0001 |
| **Clinical phenotype** | | | | |
| **Meningitis** | 542 (77.2%) | 137 (77.0%) | 405 (77.3%) | 0.93 |
| **Encephalitis** | 70 (10.0%) | 16 (9.0%) | 54 (10.3%) | 0.61 |
| **Brain abscess** | 82 (11.7%) | 21 (11.8%) | 61 (11.6%) | 0.96 |
| **Spinal disease** | 8 (1.1%) | 4 (2.3%) | 4 (0.8%) | 0.11 |
| **Aetiological agent** | | | | |
| **Bacterial** | 166 (23.7%) | 49 (27.5%) | 117 (22.3%) | 0.15 |
| **Viral** | 294 (41.9%) | 54 (30.3%) | 240 (45.8%) | <0.0001 |
| **Fungal** | 34 (4.8%) | 19 (10.7%) | 15 (2.9%) | <0.0001 |
| **No pathogen identified** | 204 (29.1%) | 56 (31.5%) | 148 (28.2%) | 0.41 |

[a] This table only presents the data of the 702 cases in whom Indigenous status was available.

**Table 3. The number of cases in Indigenous and non-Indigenous Australians by pathogen.**

| Pathogen | All n = 702[a] | Indigenous n = 178 | Non-Indigenous n = 524 | p |
|---|---|---|---|---|
| *N. meningitidis* | 34 (4.8%) | 14 (7.9%) | 20 (3.8%) | 0.03 |
| *S. pneumoniae* | 24 (3.4%) | 10 (5.6%) | 14 (2.7%) | 0.06 |
| *B. pseudomallei* | 8 (1.1%) | 2 (1.1%) | 6 (1.1%) | 0.98 |
| *M. tuberculosis* | 12 (1.7%) | 1 (0.6%) | 11 (2.1%) | 0.31 |
| *Cryptococcus species* | 33 (4.7%) | 18 (10.1%) | 15 (2.9%) | <0.0001 |
| Enterovirus | 217 (30.9%) | 45 (25.3%) | 172 (32.8%) | 0.06 |
| Herpes simplex virus-2 | 31 (4.4%) | 4 (2.2%) | 27 (5.2%) | 0.14 |
| Herpes simplex virus-1 | 18 (2.6%) | 1 (0.6%) | 17 (3.2%) | 0.06 |
| Varicella zoster virus | 14 (2.0%) | 3 (1.7%) | 11 (2.1%) | 1.0 |

[a] This table only presents the data of the 702 cases in whom Indigenous status was available.

## Meningitis

There were 561 cases of meningitis. The incidence was 8.9/100,000 population in 2000 compared to 16.7/100,000 population in 2019 (p for trend = 0.22). The patients with meningitis had a median (IQR) age of 19 (0–36) years; 302/561 (53.8%) were males. A viral aetiology was most common, with enterovirus the most frequently diagnosed. *Neisseria meningitidis* was the second most commonly identified. Cryptococcal meningitis occurred as commonly as pneumococcal meningitis (Table 5).

## Encephalitis

There were 72 cases of encephalitis. The incidence had a nadir of 0.4/100,000 population in 2003 and a peak of 2.5/100,000 population in 2018 (p for trend = 0.06); 43/72 (60%) cases occurred in males. The cases of encephalitis had a median (IQR)) age of 50 (21–66) years. Most cases had a viral aetiology, with herpes simplex virus-1 being the most common

**Table 4. The number of cases in urban and remote areas within FNQ by clinical phenotype and pathogen.**

| | All [a]n = 679 | Urban n = 599 | Remote n = 80 | p |
|---|---|---|---|---|
| **Clinical phenotype** | | | | |
| Meningitis | 529 | 469 (78.3%) | 60 (75%) | 0.50 |
| Encephalitis | 66 | 60 (10.0%) | 6 (8%) | 0.55 |
| Brain abscess | 76 | 64 (10.7%) | 12 (15%) | 0.25 |
| Spinal disease | 8 | 6 (1.0%) | 2 (2%) | 0.24 |
| **Pathogen** | | | | |
| *N. meningitidis* | 34 (5.0%) | 27 (4.5%) | 7 (9%) | 0.10 |
| *S. pneumoniae* | 22 (3.2%) | 18 (3.0%) | 4 (5%) | 0.34 |
| *B. pseudomallei* | 8 (1.2%) | 5 (0.8%) | 3 (4%) | 0.02 |
| *M. tuberculosis* | 6 (0.9%) | 4 (0.7%) | 2 (2%) | 0.10 |
| *Cryptococcus species* | 31 (4.6%) | 26 (4.3%) | 5 (6%) | 0.44 |
| Enterovirus | 216 (31.8%) | 196 (32.7%) | 20 (25%) | 0.16 |
| Herpes simplex virus-2 | 31 (4.6%) | 28 (4.7%) | 3 (4%) | 0.71 |
| Herpes simplex virus-1 | 18 (2.6%) | 18 (3.0%) | 0 | 0.12 |
| Varicella zoster virus | 14 (2.1%) | 14 (2.3%) | 0 | 0.17 |

[a] There were 679 cases with a residential address in FNQ.

**Table 5. Laboratory-confirmed aetiology of patients with meningitis.**

| Viral (n = 270) | Bacterial (n = 125) | Fungal (n = 24) |
|---|---|---|
| Enterovirus (n = 218)<br>Herpes simplex virus-2 (n = 33)<br>Varicella zoster virus (n = 12)<br>Parechovirus (n = 3)<br>Mumps (n = 2)<br>Epstein Barr virus (n = 1)<br>Human herpesvirus 6 (n = 1) | *Neisseria meningitidis* (n = 36)<br>*Streptococcus pneumoniae* (n = 24)<br>*Staphylococcus aureus* (n = 8)<br>*Streptococcus agalactiae* (n = 8)<br>*Treponema pallidum* (n = 6)<br>*Haemophilus influenzae* (n = 6)<br>*Mycobacterium tuberculosis* (n = 6)<br>*Salmonella species* (n = 5)<br>*Escherichia coli* (n = 5)<br>*Enterococcus faecalis* (n = 4)<br>*Leptospira species* (n = 3)<br>*Listeria monocytogenes* (n = 2)<br>*Streptococcus milleri* (n = 2)<br>*Burkholderia pseudomallei* (n = 1)<br>*Aeromonas hydrophilia* (n = 1)<br>*Elizabethkingia meningoseptica* (n = 1)<br>*Enterococcus faecium* (n = 1)<br>*Klebsiella pneumoniae* (n = 1)<br>*Pasteurella multocida* (n = 1)<br>*Propionibacterium acnes* [a] (n = 1)<br>*Pseudomonas aeruginosa* (n = 1)<br>*Staphylococcus epidermidis* [b] (n = 1)<br>*Streptococcus pyogenes* (n = 1) | *Cryptococcus* (n = 24)<br> • *Cryptococcus neoformans* (n = 13)<br> • *Cryptococcus gattii* (n = 7)<br> • *Cryptococcus spp* (non-speciated (n = 4) |

[a] This pathogen was isolated in a patient with a ventriculoperitoneal shunt.

[b] This pathogen was isolated in a patient with a lumbar drain.

A pathogen was not identified in 142 (25.3%) meningitis cases.

pathogen (18/72, 25%) (Table 6). There were only three cases of arboviral encephalitis, including one case of JEV imported from PNG in 2004, one case of Murray Valley encephalitis imported from PNG in 2012 and a single case of dengue. The case of dengue encephalitis was diagnosed in a local FNQ resident with no travel history during the last significant outbreak in Cairns in 2009. A case of cerebral gnathostomiasis was identified in a returning traveler from Thailand in 2019. There were 36 cases in which a pathogen was not identified; flavivirus serology was performed—and was negative—in 12 (33%) of these cases, 6 (17%) of whom also had negative PCR tests for Kunjin and Murray Valley Encephalitis. In addition to the 72 cases of encephalitis, there were 31 cases of immune-mediated encephalitis cases during the study

**Table 6. Laboratory-confirmed aetiology of patients with encephalitis.**

| Viral (n = 29) | Bacterial (n = 5) | Fungal (n = 1) | Parasitic (n = 1) |
|---|---|---|---|
| Herpes simplex virus-1 (n = 18)<br>Varicella zoster virus (n = 2)<br>JC virus [a] (n = 2)<br>Enterovirus (n = 2)<br>JEV [b] (n = 1)<br>Murray Valley encephalitis virus (n = 1)<br>Dengue (n = 1)<br>Parechovirus (n = 1)<br>Influenza A (n = 1) | *Burkholderia pseudomallei* (n = 3)<br>*Haemophilus influenzae* (n = 1)<br>*Pseudomonas stutzeri* (n = 1) | *Cryptococcus neoformans* (n = 1) | *Gnathostoma* (n = 1) |

[a] John Cunningham virus.

[b] Japanese encephalitis virus.

A pathogen was not identified in 36 (50%) encephalitis cases.

period; 9/31 (29%) were acute disseminated encephalomyelitis and 4/31 (13%) were anti-*N*-methyl-D-aspartate receptor encephalitis. Increasing numbers of cases of autoimmune encephalitis were diagnosed over the course of the study period (p for trend = 0.002).

## Brain abscess

There were 84 cases of brain abscess. The incidence was 0.9/100,000 population in 2000 compared to 3.1/100,000 population in 2019 (p for trend = 0.12). 50/84 (60%) occurred in males. The cases' median (IQR)) age was 49 (19–61) years. Tropical pathogens such as *Cryptococcus* (9/84, 11%), *Mycobacterium tuberculosis* (7/84, 8%) and *Burkholderia pseudomallei* (5/84, 6%) were as common in this cohort as more classical aetiologies (Table 7).

## Immunocompromised population

Of the 725 cases, 18 (2.5%) had confirmed human immunodeficiency virus (HIV) infection, and 20 additional cases between 2015–2019 were receiving immunosuppressive treatment. The spectrum of pathogens seen in these 38 cases was similar to those seen in temperate regions (S4 Table).

## Vaccine-preventable disease

The decline in the incidence of pneumococcal CNS infection was not statistically significant (S2 Fig). Of the 25 cases of pneumococcal CNS infection that occurred during the study period, 10 (40%) occurred after 2010, only 2 of which were caused by vaccine-preventable serotypes, 1 of whom was a 3-month-old infant (S5 Table). There was no change in the incidence of *N. meningitidis* infection during the study period (S3 Fig). There were very low

**Table 7. Laboratory-confirmed aetiology of patients with brain abscesses.**

| Bacterial (n = 45) | Fungal (n = 10) | Protozoan (n = 3) |
|---|---|---|
| *Mycobacterium tuberculosis* (n = 7)<br>*Streptococcus milleri* (n = 6)<br>*Burkholderia pseudomallei* (n = 5)<br>*Staphylococcus aureus* (n = 5)<br>*Streptococcus pyogenes* (n = 2)<br>*Nocardia paucivirans* (n = 2)<br>*Propionibacterium acnes* (n = 2)<br>*Streptococcus pneumoniae* (n = 1)<br>*Peptostreptococcus* (n = 1)<br>*Citrobacter freundii* (n = 1)<br>*Haemophilus influenzae* (n = 1)<br>*Kingella kingae* (n = 1)<br>*Listeria monocytogenes* (n = 1)<br>*Serratia marcescens* (n = 1)<br>*Staphylococcus sciuri* (n = 1)<br>*Streptococcus salivarius* (n = 1)<br>*Bacteroides* sp, anaerobic Gram negative bacilli [a]<br>*Enterococcus avium, Proteus mirabilis, Bacteroides fragilis, Bacteroides thetaiotaomicron* [a]<br>*Escherichia coli, Staphylococcus epidermidis*[a]<br>*Proteus penneri, Streptococcus pyogenes* [a]<br>*Staphylococcus aureus, Clostridium perfringens, Clostridium sordellii* [a]<br>*Streptococcus milleri, Peptostreptococcus* [a]<br>*Streptococcus sanguinis, Streptococcus parasanguinis* [a] | *Cryptococcus* (n = 9)<br>• *Cryptococcus gattii* (n = 6)<br>• *Cryptococcus neoformans* (n = 2)<br>• Non-speciated (n = 1)<br>*Aspergillus fumigatus* (n = 1) | *Toxoplasma gondii* (n = 2)<br>*Acanthamoeba* (n = 1) |

[a] These pathogens were isolated from the 7 polymicrobial brain abscesses

A pathogen was not identified in 26 (31%) brain abscess cases.

numbers of other vaccine-preventable CNS infection during the study period (8 cases of *H. influenzae*, 2 cases of mumps virus and 1 case of influenza virus).

## Management

There were 189 cases who presented to Cairns Hospital between 2015 and 2019 for whom the data on initial management was accessible (149 cases of meningitis, 18 cases of encephalitis and 22 cases of brain abscess); the care in the emergency department of 128 (67.7%) was concordant with current national therapeutic guidelines (116/149 (77.9%) cases of meningitis, 6/18 (33%) cases of encephalitis and 6/22 (27%) cases of brain abscess) [36]. The number of cases who received appropriate management in the first 24 hours rose to 161/189 (85.2%) (136/149 (91.3%) cases of meningitis, 11/18 (61%) cases of encephalitis and 14/22 (64%) cases of brain abscess). Among the 28/189 (14.8%) who didn't receive appropriate management in the first 24 hours, 8 (29%) had pathogens that were difficult to diagnose (2 cases of *M. tuberculosis*, *B. pseudomallei*, *Nocardia paucivirans* and 1 case each of *Gnathostoma* and Acanthamoeba). During the entire study period, 104/725 (14.3%) were admitted to the intensive care unit (ICU) for supportive care.

## Outcomes

Across the cohort, there were 32/725 (4.4%) deaths; the case-fatality rate remained stable over the 20-year study period (p for trend = 0.43). There were 5 deaths in infants (4 cases of meningitis and 1 case of encephalitis). All three children that died were PNG residents (2 cases of brain abscess, 1 of meningitis). Death occurred in 24 adults (12 cases of meningitis, 7 cases of encephalitis and 5 of brain abscess) (S6 Table). Death was more common in patients who had a presentation with encephalitis or a confirmed bacterial aetiology. In contrast, patients with a meningitis presentation or confirmed viral aetiology were less likely to die (Table 8). In 12/32 (38%) deaths a pathogen was not identified–this included 5 cases of meningitis, 3 cases of encephalitis and 4 cases of brain abscess. While the case-fatality rate was higher in residents of PNG than Australian residents, the difference in case-fatality rate between Indigenous and non-Indigenous patients did not reach statistical significance. The case-fatality rate was no higher in residents in remote locations (Table 8).

Among the 270 patients in whom function at discharge could be accurately determined (those managed during the from January 2015 to December 2019 period when an electronic record was used to document patient care), there were 65 with disability on discharge and 15 deaths (S7 and S8 Tables). Patients who received appropriate care in the emergency department were less likely to die or develop long term disability (odds ratio 0.11 (95% confidence interval (95% CI): 0.05–0.24), p<0.0001. In a multivariate analysis that controlled for the fact that unusual pathogens might increase the risk of poor outcomes, the risk of death or long-term disability was still lower among those who received appropriate care in the emergency department (OR (95%CI): 0.13 (0.06–0.30), p<0.0001).

## Discussion

In this region of tropical Australia, the annual incidence of CNS infection due to tropical pathogens such as *B. pseudomallei* and *Cryptococcus spp*. is similar to that of traditional bacterial pathogens. The local Aboriginal and Torres Strait Islander people bear a disproportionate burden of CNS infections, with cryptococcal and meningococcal disease seen more commonly in these populations. However, residents of remote areas were not over-represented, despite the potentially increased risk of exposure to vectors and pathogens in these locations [27]. Pathogens reported frequently from tropical regions in Southeast Asia—including JEV, dengue and

**Table 8. The number of cases in those who survived and died, stratified by clinical phenotypes, demographic characteristics and pathogens.**

|  | All n = 725 | Survived n = 693 | Died n = 32 | p |
|---|---|---|---|---|
| **Clinical phenotypes** | | | | |
| **Meningitis** | 561 (77.4%) | 544 (78.5%) | 17 (53%) | 0.001 |
| **Encephalitis** | 72 (9.9%) | 64 (9.2%) | 8 (25%) | 0.004 |
| **Brain Abscess** | 84 (11.6%) | 77 (11.1%) | 7 (22%) | 0.08 |
| **Spinal disease** | 8 (1.1%) | 8 (1.1%) | 0 | 1.0 |
| **Aetiological agent** | | | | |
| **Bacterial** | 175 (24.1%) | 162 (23.4%) | 13 (41%) | 0.03 |
| **Viral** | 299 (41.2%) | 296 (42.7%) | 3 (9%) | <0.0001 |
| **Fungal** | 35 (4.8%) | 31 (4.5%) | 4 (12%) | 0.06 |
| **Not identified [a]** | 212 (29.2%) | 200 (28.9%) | 12 (38%) | 0.30 |
| **Demographic characteristics** | | | | |
| **Infant** | 176 (24.3%) | 171 (24.7%) | 5 (16%) | 0.30 |
| **Child** | 124 (17.1%) | 121 (17.5%) | 3 (9%) | 0.34 |
| **Adult** | 425 (58.6%) | 401 (57.9%) | 24 (75%) | 0.06 |
| **Indigenous Australian [b]** | 178/702 (25.4%) | 168/675 (24.9%) | 10/27 (37%) | 0.16 |
| **Remote residence** | 80/679 (11.8%) | 77/654 (11.8%) | 3/25 (12%) | 1.0 |
| **Papua New Guinean residents** | 32 (4.4%) | 28 (4.0%) | 4 (12%) | 0.047 |
| **Pathogen** | | | | |
| *N. meningitidis* | 36 (5.0%) | 35 (5.0%) | 1 (3%) | 1.0 |
| *S. pneumoniae* | 25 (3.4%) | 25 (3.6%) | 0 | 0.62 |
| *S. aureus* | 13 (1.8%) | 10 (1.4%) | 3 (9%) | 0.03 |
| *M. tuberculosis* | 13 (1.8%) | 11 (1.6%) | 2 (6%) | 0.11 |
| *B. pseudomallei* | 9 (1.2%) | 7 (1.0%) | 2 (6%) | 0.06 |
| *Cryptococcus species* | 34 (4.7%) | 30 (4.3%) | 4 (12%) | 0.06 |
| **Enterovirus** | 220 (30.3%) | 220 (31.7%) | 0 | <0.0001 |
| **HSV1** | 18 (2.5%) | 16 (2.3%) | 2 (6%) | 0.19 |
| **Management** | | | | |
| **Appropriate therapy in the emergency department [c]** | 128/189 (67.7%) | 127/185 (68.6%) | 1/4 (25%) | 0.10 |
| **Appropriate therapy in first 24 hours [c]** | 161/189 (85.2%) | 158/185 (85.4%) | 3/4 (75%) | 0.48 |

[a] 4 other pathogens (2 toxoplasma, 1 acanthamoeba, 1 gnathostomiasis) all survived.

[b] This includes the 702 cases in whom Indigenous status was available.

[c] Only the 189 patients managed at Cairns Hospital between 2015 and 2019 for whom complete data were available.

leptospirosis [37,38]—were very rare causes of CNS disease in FNQ, highlighting the success of local public health strategies in preventing these infections. There were also low rates of vaccine-preventable disease—especially influenza, measles, and mumps—with the incidence of invasive pneumococcal and meningococcal disease projected to decline further with recent changes in the national immunisation schedule [39–43]. Despite the region's proximity to PNG—and Cairns' reputation as an international travel hub—the burden of imported CNS infection was very low.

*Cryptococcus spp*. and *B. pseudomallei* were seen as commonly as traditional bacterial pathogens such as *S. pneumoniae* and *N. meningitidis*. It is important for local clinicians to be aware of this fact as CNS cryptococcosis and melioidosis may be rapidly fatal in the absence of early appropriate treatment; if the diagnoses are not considered, empirical regimens are unlikely to contain effective therapy [3,44]. Cryptococcal CNS infection was more common in the Indigenous population, echoing findings from a previous Australasian study [45]. In that

study this was attributed to Indigenous Australians' socioeconomic disadvantage, their higher prevalence of comorbidities and their greater exposure to the Eucalyptus trees (an environmental reservoir for *Cryptococcus gattii*) in the rural areas where they comprise a greater proportion of the population. The incidence of melioidosis is increasing in FNQ, and although CNS involvement is infrequent, a high index of suspicion is necessary in the appropriate clinical context [3,10,46,47]. Other tropical pathogens were not encountered as frequently; there were only three cases of leptospiral meningitis and no rickettsial CNS infections in this study, despite FNQ having the highest incidence of leptospirosis in Australia and a rising incidence of rickettsial infections [2,48].

Indigenous Australians bore a disproportionate burden of CNS infection; over 25% of the local residents in the cohort identified as Indigenous, compared with 17% of the local general population. However, in contrast to other infectious diseases in the region, the difference in CNS infection case-fatality rates between the Indigenous and non-Indigenous population did not reach statistical significance [12,49]. The absence of a difference in the case-fatality rate may be partly explained by the fact that Indigenous Australians were less likely to be diagnosed with viral CNS infection, particularly lower virulence pathogens like enterovirus and HSV-2. This may reflect a higher threshold for performing lumbar punctures in the remote locations in which Indigenous Australians represent a greater proportion of the population. Indigenous infants, in particular, were over-represented in this study, with disproportionately high rates of meningococcal infection, highlighting the ongoing need for optimal vaccine coverage and effective, practical strategies to improve the socioeconomic disadvantage affecting many Indigenous Australians [14,24,39,50,51]. Although the meningococcal vaccine can only be provided after 6 weeks of age, immunising household contacts of infants may prevent invasive meningococcal disease in this age group [52].

*S. pneumoniae*, *N. meningitidis* and *H. influenzae* type b (Hib) were important causes of vaccine-preventable CNS infection in the cohort. The incidence of invasive pneumococcal disease across Australia declined by up to 39% across all ages after adding the 7-valent conjugated pneumococcal vaccine and the pneumococcal polysaccharide vaccine to the National Immunisation Program in 2005 and 13-valent conjugated pneumococcal vaccine in 2011 [39,53]. Although, in this small study, the overall decline in the incidence of pneumococcal CNS infection was not statistically significant, almost 96% of 5-year-old children in the region are fully vaccinated and this may explain why there were only two cases of pneumococcal CNS infection caused by vaccine-preventable serotypes in the region after 2011 [54]. The only child to have pneumococcal CNS infection with a vaccine-preventable serotype after 2011 was 3 months old, not old enough to have completed their vaccination schedule. The Australian incidence of invasive meningococcal serogroup C declined 96% in the 10 years after the introduction of the meningococcal C conjugate vaccine in 2003 [55]. There were numerically fewer cases of meningococcal CNS infection in the last ten years—despite the growing population—however, as the ACWY vaccine was only added to the schedule in 2017 and as data on meningococcal serotypes was not collected in this study, it is not possible to comment on trends in local incidence. Introduction of the Hib vaccine in 1992 explains the low rates of this infection in the cohort [16]. Despite the region's proximity to PNG—where the incidence of subacute sclerosing panencephalitis is high and vaccine coverage is suboptimal [56], there were no cases of measles-related encephalitis in the study. There were only two cases of mumps encephalitis during the study period locally (both occurring in adults), likely related to the high local rates of vaccine coverage in FNQ [54].

Arbovirus vector control interventions, including mosquito surveillance, population control and the introduction of the *Wolbachia* program, have been extremely successful in reducing the burden of mosquito-borne disease [20,21]. There has been only one local case of

dengue encephalitis—which occurred during the last major outbreak of dengue in the region in 2009—and one imported case of JEV in the last 20 years [57,58]. There have been no locally acquired dengue CNS infections since deployment of *Wolbachia*-infected mosquitoes began in 2011, despite importation of dengue in travellers, particularly those returning from the Asia-Pacific region [6,21]. This stands in stark contrast to some Southeast Asian locations where flavivirus infections are responsible for a significant proportion of all CNS infections [37,38,58]. Incursions of JEV into FNQ occurred in 1998, 2000 and 2004, when infected mosquitoes were carried by cyclonic winds from PNG to the Torres Strait Islands and Cape York Peninsula. Although the virus was identified almost annually in sentinel pigs and mosquitoes on Badu Island in the Torres Strait until the discontinuation of these surveillance programs in 2005, there were no locally acquired cases in the study period; vaccination for residents of the outer Torres Strait Islands may have played a role [6,26]. The sole case of JEV was detected in an expatriate who was aeromedically evacuated from PNG in 2004 [57]. Despite Murray Valley encephalitis virus and Kunjin virus being endemic to northern Australia, there were no locally acquired cases of these flaviviruses diagnosed in this cohort with the only case of Murray Valley encephalitis occurring in a PNG resident. Recent cases of these flaviviruses appear confined to Western Australia and Northern Territory where rainfall conditions are optimal for vector survival and breeding [59,60]. Although emerging pathogens including Australian bat lyssavirus and Hendra virus are highlighted in the national consensus guidelines for encephalitis in Australia [27]—and have been identified in Queensland [61,62]—no cases were identified in the 20 years of this study.

Although tuberculosis was the most commonly identified CNS infection in PNG residents, it was identified in only a single Aboriginal or Torres Strait Islander Australian in the study period. Declining rates of tuberculosis have been noted among local Indigenous Australians following improved engagement with Indigenous communities, training of local healthcare workers and an enhanced specialist outreach program [63]. PNG is one of the world's 30 high burden TB countries suggesting that early detection and contact tracing strategies in the Torres Strait are currently limiting the importation of TB into FNQ [6].

The Western Pacific region has seen a 43% decline in cases of malaria, but nearly 80% of cases in the region are reported from PNG [30]. It is notable then, that despite PNG's proximity, the number of cases of *P. falciparum* malaria seen in FNQ is decreasing. Only 10.8% of all cases from the study period occurred in the second half of the study, only one of whom–a PNG resident–had impaired consciousness. Apart from tuberculosis, malaria, and the 2 cases of JEV and Murray Valley encephalitis, the other tropical pathogens seen in PNG residents were similar to those encountered in northern Australia.

The case-fatality rate in the study was far lower than that seen in contemporary international series from tropical locations [37,38] (S9 Table). This likely reflects management in the well-resourced Australian public health system in which patients are able to access care in even remote locations, and have access to efficient aeromedical retrieval service, comprehensive diagnostic services—including advanced imaging—and sophisticated ICU and neurosurgical support [47,64,65]. A pathogen was identified in 71.1% of cases, a larger proportion than that seen in a study examining encephalitis in Australia between 1979 and 2006 [66], although this is likely due to the use of ICD coding—and the pathology database—to identify cases. The difference in the spectrum of isolated pathogens—especially the greater proportion of lower virulence pathogens such as enterovirus seen in the FNQ cohort—is also likely to have contributed to the lower case-fatality rate. Although the proportion of encephalitis deaths from unidentified pathogens was reported to be high in a previous Australian study [66], there were only 12 deaths due to CNS infection from unidentified pathogens over 20 years in this series–and only 3 of these were due to encephalitis. It was notable that an autoimmune aetiology explained

approximately a third of encephalitis presentations with the incidence increasing during the study period, at least in part due to greater local recognition of the syndrome [67]. Future research employing novel molecular diagnostic techniques may help identify emerging infections, guide targeted therapy, and further inform optimal management strategies [68].

This study was limited by incomplete access to medical records and imaging which precluded comprehensive data collection. The incidence of individual pathogens was relatively low increasing the likelihood of a type 2 statistical error. The Modified Rankin Scale score and requirement of an anticonvulsant medication on discharge are crude proxy measures of long-term disability [34]. Using ICD coding and searching the laboratory database for any CSF that had identified an organism may have resulted in a lower number of cases with an unidentified pathogen. Additionally, cases of mild, self-limiting CNS infection may have been missed in remote locations due to a higher threshold for performing lumbar punctures in these settings. There was no standardised testing of patients with each of the clinical syndromes and testing was not always comprehensive; it is therefore possible that the failure to identify individual pathogens—particularly flaviviruses, Hendra virus and Australian bat lyssavirus—was the result of not testing for them. Different methods used by the Australian Bureau of Statistics to define the FNQ population led to minor variations in the size of the local population in the analyses that compared characteristics of the cohort to that of the general population, although this would not be expected to influence the study's findings. Despite these limitations, this study has identified the evolving spectrum of pathogens that are responsible for CNS infections in the region. Just as importantly, it highlights the pathogens that—despite being emphasised in national consensus statements—are actually identified infrequently.

## Conclusion

In this region of Australia, tropical pathogens caused CNS infections as frequently as classical bacterial pathogens, however, only one case of locally acquired arboviral encephalitis occurred over the 20 years of the study. The case-fatality rate was very low compared to contemporary series from Southeast Asia, reflecting differences in the array of responsible pathogens and Australia's well-resourced health care system. With effective vector control, expanding comprehensive vaccination schedules and an emphasis on prompt diagnosis and therapy, death, and disability from CNS infections in the region is expected to decline even further in the future.

## Supporting information

**S1 Fig. Cases of CNS infection satisfying the criteria for inclusion in the study.**
(TIF)

**S2 Fig. *S. pneumoniae* incidence per 100,000 population.**
(TIF)

**S3 Fig. *N. meningitidis* overall incidence per 100,000.**
(TIF)

**S1 Table. Incidence per 100,000 local population between 2000 and 2019.**
(DOCX)

**S2 Table. Pathogens identified in Aboriginal and Torres Strait Islander children and infants.**
(DOCX)

**S3 Table. Pathogens in patients who lived outside Australia.**
(DOCX)

**S4 Table. Pathogens identified in immunocompromised patients.**
(DOCX)

**S5 Table. Serotypes identified in pneumococcal CNS infection.**
(DOCX)

**S6 Table. Pathogens causing CNS infection in patients who died.**
(DOCX)

**S7 Table. Death and disability outcomes stratified by clinical phenotypes, aetiology and demographic characteristics.**
(DOCX)

**S8 Table. Residual deficits in infants, children and adults after CNS infection, and associated pathogens.**
(DOCX)

**S9 Table. Most common pathogens causing CNS infection and case fatality rates in Far North Queensland, Vietnam and Laos [4,5].**
(DOCX)

**S1 File. Case definitions of CNS infections.**
(DOCX)

**S2 File. Case definitions of autoimmune encephalitis.**
(DOCX)

**S1 Dataset.**
(XLSX)

## Acknowledgments

We thank Peter Horne (Cairns Tropical Public Health Unit) for his assistance with the production of Fig 1.

## Author Contributions

**Conceptualization:** Simon Smith, Ian Wilson, Josh Hanson.

**Data curation:** Hannah Gora, Simon Smith, Josh Hanson.

**Formal analysis:** Hannah Gora, Josh Hanson.

**Investigation:** Hannah Gora, Simon Smith, Ian Wilson, Josh Hanson.

**Methodology:** Simon Smith, Josh Hanson.

**Software:** Josh Hanson.

**Supervision:** Simon Smith, Ian Wilson, Josh Hanson.

**Validation:** Annie Preston-Thomas, Nicole Ramsamy, Josh Hanson.

**Visualization:** Hannah Gora, Josh Hanson.

**Writing – original draft:** Hannah Gora.

**Writing – review & editing:** Hannah Gora, Simon Smith, Ian Wilson, Annie Preston-Thomas, Nicole Ramsamy, Josh Hanson.

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
