## [Decision Letter · Decision Letter 0]

5 Jan 2022

PONE-D-21-35660The epidemiology and outcomes of central nervous system infections in Far North Queensland, tropical Australia; a 20-year retrospective studyPLOS ONE

Dear Dr. Gora,

Thank you for submitting your manuscript to PLOS ONE. After careful consideration, we feel that it has some merit but does not fully meet PLOS ONE’s publication criteria as it currently stands. Therefore, we invite you to submit a  revised version of the manuscript that addresses the points raised during the review process. Please pay particular attention to the confusing way numbers are presented in this manuscript.  Especially, please go through the choice of denominator for calculation of each frequencies (ie morbidity and mortality statistics).  Please tighten the paper by increasing cohesiveness, focus and clarity.  Reduce the number of tables to absolute necessity. Please make sure to address ALL comments from the reviewers before uploading the next version of the manuscript.

We look forward to receiving your revised manuscript.

Kind regards,

Latiffah Hassan

Academic Editor

PLOS ONE

Journal Requirements:

2. We note that Figure 1 in your submission contain map images which may be copyrighted. All PLOS content is published under the Creative Commons Attribution License (CC BY 4.0), which means that the manuscript, images, and Supporting Information files will be freely available online, and any third party is permitted to access, download, copy, distribute, and use these materials in any way, even commercially, with proper attribution. For these reasons, we cannot publish previously copyrighted maps or satellite images created using proprietary data, such as Google software (Google Maps, Street View, and Earth). For more information, see our copyright guidelines: http://journals.plos.org/plosone/s/licenses-and-copyright.

Reviewers' comments:

Reviewer's Responses to Questions

**Comments to the Author**

1. Is the manuscript technically sound, and do the data support the conclusions?

Reviewer #1: Yes

Reviewer #2: Partly

2. Has the statistical analysis been performed appropriately and rigorously? 

Reviewer #1: Yes

Reviewer #2: Partly

3. Have the authors made all data underlying the findings in their manuscript fully available?

Reviewer #1: Yes

Reviewer #2: Yes

4. Is the manuscript presented in an intelligible fashion and written in standard English?

Reviewer #1: Yes

Reviewer #2: Yes

5. Review Comments to the Author

Reviewer #1: Overall, this is an interesting study detailing changes in the epidemiology of CNS infections in a region of Australia over the past 20 years.

Specific comments are as follows:

INTRODUCTION

- lines 80-82, there are probably additional reasons for a higher prevalence of some infectious diseases in PNG compared to Australia than the “fragile public health system.” I would just state that PNG has a higher prevalence of these infections and leave it at that.

- line 100 – I would specify here that Wolbachia blocks the insects ability to spread dengue, Zika, and chikungunya as I don’t think the average reader would be aware of this.

METHODS

- Lines 142-143 would specify that collected data were de-identified

- Lines 142-143 – the way this is worded is really confusing because earlier they mention that patients between 2000 and 2019 are included. I would re-word this to state that for the subset of patients between 2015 and 2019 additional clinical information, such as demographics etc. was collected.

RESULTS

- Lines 201-202 – it says Cryptococcal and meningococcal infection were more common in Indigenous Australians... Would add “compared to non-indigenous Australians” to that sentence.

- Lines 222-223 – I would also show the one case of TB in non-PNG individuals (with denominator) here, to highlight how different this was between groups.

- Lines 223-224 The sentence that starts with “there were 15 other patients in the cohort ...“ doesn’t really fit here. It should be deleted or moved to a different part of the paper.

- Line 235 – Even though the authors state earlier that they used CDC case definition of meningitis, I would add a sentence to clarify how they diagnosed these cases of meningitis – did they all have lumbar punctures, or were any of them diagnosed based on clinical symptoms only?

- Lines 235-240 – Were there any differences between immunocompromised and non-immunocompromised individuals? These might be interesting data to include here.

- Lines 277-280 – Were there any differences in the spectrum of diseases (meningitis vs. encephalitis) among immunocompromised and non-immunocompromised patients?

- Lines 303-304 – I would list admission to ICU as an outcome, not a treatment.

- Lines 322-323 – They say 65 had disability on discharge and “an additional 15 deaths” – does this mean 15 more subjects died following discharge? This statement is confusing.

DISCUSSION

- Lines 339-341 – They say residents of remote areas were not over-represented, despite increased risks of exposure to vectors. However, I would think it would be likely that those in remote areas may be less likely to seek care.

- Lines 354-358 – was there any information regarding the association between HIV status and cryptococcal diagnosis

- Lines 418-420 – they state that Murray Valley Encephalitis virus and Kunjin virus are endemic but there were no cases of these diagnosed – in how many cases were these viruses tested for??

TABLES

- Table 1 – for the descriptive statistics shown in the tables, I’m not sure it makes sense to include p-values. However, if you do include them (as is shown in Tables 2 and 3) they should also be listed in Table 1 then.

- Table 5 – were there no cases that had >1 pathogen?? If there were, this should be clarified in a footnote.

- Table 9 – I would not include this whole table in the discussion. You could make it a supplemental table and refer to it in text, but typically tables are not included in the discussion.

Reviewer #2: The authors try to highlight the epidemiological profiles and clinical outcomes of patients who were diagnosed with central nervous system infections in Far North Queensland, tropical Australia. The data were retrospectively collected from 1st January 2000 until 31st December 2019. I find the manuscript is too verbose and lack of novelty. The abstract was not comprehensively written. For example, the aim of the study was not properly written in the background with lack of problem statement. Methodology was not very clear in terms of the lab tests/methods used for the identification of the pathogens.

Several variables should be defined as follows: tropical pathogens, indigenous communities, not identified pathogens (Table 1), Indigenous Australians, Local Indigenous, disability outcomes, and residual deficits.

Why the first letter of “Indigenous” is capitalized, especially when it is written as Indigenous communities or Indigenous status.

Methods:

The inclusion and exclusion criteria were not clearly mentioned. Were repeat samples included? Were patients with coinfections included?

Line 142-143: “The patients’ demographics, clinical features, comorbidities, and outcomes were collected from the medical records of cases between January 1, 2015, and December 31, 2019” There is a big gap on the data collected between 2000 to 2014, which may affect the statistical analysis in general.

Line 151 -153: the statement should be cited.

Results:

The analysis was affected by different denominators used and this has created a lot of confusion to the readers.

Table 2: what is the purpose of comparing indigenous versus non-indigenous? This was not highlighted in the introduction.

Species should not be italicised.

About enterovirus, was it only one genus identified? What method was used for the identification?

There were too many tables provided, which I think some are not necessary especially the impact of residential address. It would be good if geographic information system is utilized.

Line 312-313: statistical results should be provided.

Discussion:

Line 379: Spelling error was observed, N. meningitides

Table 9 should be de-tabulated, and the authors should use all the points for the discussion. The table is more appropriate for a review paper.

Epidemiological parameters were not critically discussed in relation to CNS infections. Lack of references for the comparison was observed.

6. PLOS authors have the option to publish the peer review history of their article (what does this mean?). If published, this will include your full peer review and any attached files.

Reviewer #1: **Yes: **Jesica A Herrick

Reviewer #2: No

---

## [Author Response · Author response to Decision Letter 0]

19 Jan 2022

Response to Reviewers

We thank the editorial staff and reviewers for the time that they have taken to review our manuscript and the helpful comments that they have made to improve the work. Please find below our point-by-point responses to their comments. For clarity, the editorial staff and reviewers’ comments are presented in blue normal text, while our responses are in black normal text. Any modified – or highlighted – text is presented as black and italicized.

Editorial staff comments

Response: We have checked that the manuscript meets the style requirements – including file naming – and feel that it does. We are very happy to address specific issues if any remain. 

2. We note that Figure 1 in your submission contain map images which may be copyrighted. All PLOS content is published under the Creative Commons Attribution License (CC BY 4.0), which means that the manuscript, images, and Supporting Information files will be freely available online, and any third party is permitted to access, download, copy, distribute, and use these materials in any way, even commercially, with proper attribution. For these reasons, we cannot publish previously copyrighted maps or satellite images created using proprietary data, such as Google software (Google Maps, Street View, and Earth). For more information, see our copyright guidelines: http://journals.plos.org/plosone/s/licenses-and-copyright.

We require you to either (1) present written permission from the copyright holder to publish these figures specifically under the CC BY 4.0 license, or (2) remove the figures from your submission.

Response: The map in the submission was created using mapping software (MapInfo version 15.02, Connecticut, USA) using data provided by the State of Queensland (QSpatial) which are not copyrighted. Queensland Place Names — State of Queensland (Department of Natural Resources, Mines and Energy) 2019, available under Creative Commons Attribution 4.0 International licence https://creativecommons.org/licenses/by/4.0/ and ‘Coastline and state border–Queensland - State of Queensland (Department of Natural Resources, Mines and Energy) 2019, available under Creative Commons Attribution 4.0 International licence https://creativecommons.org/licenses/by/4.0/

These data - which are freely available in the public domain - are provided by the State Government of Queensland. This is similar to some of the U.S. government sources that were suggested in the decision email. 

All the patients were managed in the State Government’s Public Health system. All the authors contributed to the manuscript while working or studying in the State Government’s Public Health system.

We have used these data to create maps in multiple PLoS publications in the past 4 years without any issue. 

1. PLoS Negl Trop Dis. 2021 Jun 21;15(6):e0009544. doi: 10.1371/journal.pntd.0009544. eCollection 2021 Jun.

2. PLoS Negl Trop Dis. 2021 Jan 14;15(1):e0008990. doi: 10.1371/journal.pntd.0008990. eCollection 2021 Jan.

3. PLoS One. 2020 Sep 3;15(9):e0238719. doi: 10.1371/journal.pone.0238719. eCollection 2020.

4. PLoS Negl Trop Dis. 2019 Jul 18;13(7):e0007583. doi: 10.1371/journal.pntd.0007583. eCollection 2019 Jul.

5. PLoS Negl Trop Dis. 2019 Feb 13;13(2):e0007205. doi: 10.1371/journal.pntd.0007205. eCollection 2019 Feb.

6. PLoS Negl Trop Dis. 2017 Mar 6;11(3):e0005411. doi: 10.1371/journal.pntd.0005411. eCollection 2017

We have added a footnote to Figure 1 – specifically addressing this copyright issue – to make this clearer. Hopefully this addresses your concerns.

REVIEWERS COMMENTS

Reviewer #1: Overall, this is an interesting study detailing changes in the epidemiology of CNS infections in a region of Australia over the past 20 years.

Specific comments are as follows:

INTRODUCTION

- lines 80-82, there are probably additional reasons for a higher prevalence of some infectious diseases in PNG compared to Australia than the “fragile public health system.” I would just state that PNG has a higher prevalence of these infections and leave it at that.

Response: We agree with the reviewer, the original description was too simplistic. We have deleted the phrase as suggested (lines 80-81).

- line 100 – I would specify here that Wolbachia blocks the insects ability to spread dengue, Zika, and chikungunya as I don’t think the average reader would be aware of this.

Response: We thank the reviewer for highlighting this important point. We have amended the paper accordingly (line 104-105). 

METHODS

- Lines 142-143 would specify that collected data were de-identified

Response: We agree that this is important. We did state that the data were de-identified in the statistics section of the original submission (line 189).

- Lines 142-143 – the way this is worded is really confusing because earlier they mention that patients between 2000 and 2019 are included. I would re-word this to state that for the subset of patients between 2015 and 2019 additional clinical information, such as demographics etc. was collected.

Response: We thank the reviewer for highlighting this source of potential confusion. We have revised the paper accordingly (lines 158-162).

RESULTS

- Lines 201-202 – it says Cryptococcal and meningococcal infection were more common in Indigenous Australians... Would add “compared to non-indigenous Australians” to that sentence.

Response: We thank the reviewer for this helpful suggestion. We have revised the paper accordingly (lines 229-230).

- Lines 222-223 – I would also show the one case of TB in non-PNG individuals (with denominator) here, to highlight how different this was between groups.

Response: This is an excellent suggestion. We have revised the paper accordingly (line 253-254).

- Lines 223-224 The sentence that starts with “there were 15 other patients in the cohort ...“ doesn’t really fit here. It should be deleted or moved to a different part of the paper.

Response: We thank the reviewer for their constructive comment. We have revised the paper accordingly (line 254). 

- Line 235 – Even though the authors state earlier that they used CDC case definition of meningitis, I would add a sentence to clarify how they diagnosed these cases of meningitis – did they all have lumbar punctures, or were any of them diagnosed based on clinical symptoms only?

Response: We agree with the reviewer that the method of diagnosing CNS infections is important. This is described in detail in Supplementary File 1. 

- Lines 235-240 – Were there any differences between immunocompromised and non-immunocompromised individuals? These might be interesting data to include here.

- Lines 277-280 – Were there any differences in the spectrum of diseases (meningitis vs. encephalitis) among immunocompromised and non-immunocompromised patients?

Response: We agree with the reviewer that immunocompromise is an important variable worth examining. The difference in case-fatality rate between immunocompromised and non-immunocompromised individuals did not reach statistical significance (p=0.14) and we list all the pathogens that cause infection in the immunocompromised patients in Supplementary Table 4.

However, detailed discussion of the differences in individual infections between immunocompromised (who represented only 2.5% of the cohort) and non-immunocompromised patients is beyond the scope of this paper which instead aimed to focus on long term trends and the implications for public health strategy in the region. Two manuscripts - one examining Cryptococcal infection and another examining CNS melioidosis in the region during the same time period - have been submitted to academic journals and are currently under review.

- Lines 303-304 – I would list admission to ICU as an outcome, not a treatment.

Response: The reviewer is right to note that some studies use ICU admission as an outcome. However, we feel that it is better placed in the management paragraph, as this is how patients were managed by the attending clinicians. This means that the more traditional long-term outcomes - death and long-term disability can be discussed alone in the outcome section. 

- Lines 322-323 – They say 65 had disability on discharge and “an additional 15 deaths” – does this mean 15 more subjects died following discharge? This statement is confusing.

Response: We thank the reviewer for highlighting this source of potential confusion. 15 of the 32 total deaths occurred between January 2015 and December 2019. We have revised the paper accordingly to clarify this (line 360). 

DISCUSSION

- Lines 339-341 – They say residents of remote areas were not over-represented, despite increased risks of exposure to vectors. However, I would think it would be likely that those in remote areas may be less likely to seek care.

Response: We agree with the reviewer that cases of mild CNS infection may have been missed in residents of remote areas. Indeed, we present data to show that lumbar punctures were performed less frequently in rural and remote settings (lines 248-251). This point has been specifically acknowledged as a limitation of the study (lines 509-510). 

- Lines 354-358 – was there any information regarding the association between HIV status and cryptococcal diagnosis

Response: There were only 3 cases of Cryptococcal disease in HIV seropositive patients during the entire study period. The association between HIV infection and cryptococcal disease is well recognised and it is probably unnecessary to describe it in more detail.

As highlighted above, we have another paper under review at PLoS One examining Cryptococcal infections in the region where we discuss the issue in more detail (PONE-D-21-37969R1. The aetiology and clinical characteristics of cryptococcal infections in Far North Queensland, tropical Australia). 

There were 18 episodes of CNS infection in HIV seropositive patients during the study period. This included 3 cases of Cryptococcal disease, 2 cases of Toxoplasmosis gondii, 1 case of JC virus, 1 case of TB, 1 case of Salmonella yarrabah, 1 case of Listeria monocytogenes, 1 case of Human Herpes Virus 6, 1 case of Herpes Zoster and 7 cases in which the pathogen was not identified. 

These pathogens are all presented in supplementary table 4 and further detailed description of infections in this relatively small subgroup (approximately 2.5% of the cohort) is probably beyond the scope of the manuscript, however we would be happy to expand this section if the Editor thought that it would be valuable.

- Lines 418-420 – they state that Murray Valley Encephalitis virus and Kunjin virus are endemic but there were no cases of these diagnosed – in how many cases were these viruses tested for??

Response: The reviewer makes an excellent point and one that we should have presented in the initial submission. There were 36 cases of encephalitis in which no pathogen was identified. Flavivirus serology was performed - and was negative - in 12. PCR for Kunjin and Murray Valley Encephalitis was performed - and was negative - in 6. It is therefore possible that cases of either pathogen were missed in the remaining 24 cases; 2 of these cases died. We have expanded the results to highlight this point (lines 291-293) and we have also added this as a limitation of the study (lines 511-514).

TABLES

- Table 1 – for the descriptive statistics shown in the tables, I’m not sure it makes sense to include p-values. However, if you do include them (as is shown in Tables 2 and 3) they should also be listed in Table 1 then.

Response: There are no comparisons - or analyses - made in Table 1 so there are no p-values to present. We could perform an analysis, but differences in CNS infection between adults, children and infants have been described previously. Furthermore, it is more complicated to present statistical analyses between three groups succinctly. The absolute numbers presented in a descriptive manner perhaps tell their own story: meningitis occurs more commonly in infants; fungal pathogens are seen almost exclusively in adults. 

This is different from Tables 2 and 3 which compare the number of cases in Indigenous and non-Indigenous Australians - using the chi-square or Fisher’s exact test - in each of the categories. There is very little data that compares the burden of CNS infections in Indigenous and non-Indigenous Australians, so we have performed a statistical analysis and presented the p values accordingly.

If the Editor feels that the paper would be stronger performing analyses comparing the incidence of the different clinical phenotypes and pathogens in adults, children and infants, we would be happy to present this.

- Table 5 – were there no cases that had >1 pathogen?? If there were, this should be clarified in a footnote.

Response: There were no cases of meningitis or encephalitis with more than one pathogen identified. Cases of polymicrobial brain abscesses are indicated by a superscript a in Table 7, as detailed in the footnote (line 313). 

- Table 9 – I would not include this whole table in the discussion. You could make it a supplemental table and refer to it in text, but typically tables are not included in the discussion.

Response: We thank the reviewer for their constructive comment. We have relegated the table to supplementary material (supplementary table 9).

Reviewer #2: The authors try to highlight the epidemiological profiles and clinical outcomes of patients who were diagnosed with central nervous system infections in Far North Queensland, tropical Australia. The data were retrospectively collected from 1st January 2000 until 31st December 2019. I find the manuscript is too verbose and lack of novelty. The abstract was not comprehensively written. For example, the aim of the study was not properly written in the background with lack of problem statement. 

Response: We agree with the reviewer about the importance of a clear and concise abstract. Our abstract is 294 words (6 words less than the word limit set by PLOS ONE). We would argue that we do present the aim of the study in the abstract: to define the epidemiology of central nervous system infections in Far North Queensland (lines 27-28).

We present the aims of the study, in some detail, in the introduction (as per STROBE guidelines) (lines 113-121).

Methodology was not very clear in terms of the lab tests/methods used for the identification of the pathogens.

Response: We agree with the reviewer that it is important to define the laboratory methods used to identify the pathogens. However, given the large number of pathogens assayed using a variety of different tests, we felt that describing the laboratory tests in further detail would impact significantly on the length and legibility of the article. Cryptococcal infection, for instance, may be identified by India Ink stain, antigen testing (of CSF or serum) or culture. Leptospirosis might be identified by serology, PCR, or culture. These are just two of the pathogens that were sought, and we feel that detailing all these tests for the large variety of pathogens would be unnecessarily inclusive. We would be happy to expand the current description (lines 133-134) if the Editor felt that this were appropriate. 

Several variables should be defined as follows: tropical pathogens, indigenous communities, not identified pathogens (Table 1), Indigenous Australians, Local Indigenous, disability outcomes, and residual deficits.

Response: We thank the reviewer for highlighting this oversight. We have revised the paper accordingly to define Aboriginal and Torres Strait Islander Australians (lines 162-164). ‘Not identified pathogens’ includes cases where a pathogen was not identified by culture, PCR, antigen, or antibody detection. We feel that this definition is implied and unnecessary to define further, however we would be happy to do so if the Editor felt that this would improve the legibility of the text. We have added a definition for tropical pathogens in the methods (lines 180-181), however in almost every reference to tropical pathogens we do refer to the pathogens that we are discussing (lines 44-46, lines 306-308, lines 374-375, lines 399-402). We define disability in the original manuscript in lines 181-183.

Why the first letter of “Indigenous” is capitalized, especially when it is written as Indigenous communities or Indigenous status.

Response: The Australian Bureau of Statistics Standards for Statistics on Cultural and Language Diversity state that the word ‘Indigenous’ should always be capitalised, as capitalisation demonstrates respect. 

Methods: The inclusion and exclusion criteria were not clearly mentioned. Were repeat samples included? Were patients with coinfections included?

Response: We thank the reviewer for highlighting this oversight. Repeat samples were not included, we have revised the paper to reflect this (line 135). The few patients with coinfections - all brain abscesses - have this highlighted in table 7. 

Line 142-143: “The patients’ demographics, clinical features, comorbidities, and outcomes were collected from the medical records of cases between January 1, 2015, and December 31, 2019” There is a big gap on the data collected between 2000 to 2014, which may affect the statistical analysis in general.

Response: We completely agree with the reviewer that incomplete data collection and the inclusion of a small dataset may contribute to a type 2 statistical error. This is acknowledged as a limitation of the study (lines 504-505). 

Line 151 -153: the statement should be cited.

Response: We thank the reviewer for highlighting this oversight. We have revised the paper accordingly (reference 32). 

Results: The analysis was affected by different denominators used and this has created a lot of confusion to the readers.

Response: We agree with the reviewer that there are various denominators used; this was unfortunately unavoidable given the long study period and the sometimes incomplete data available for this retrospective study. We have done our best to indicate the denominators, always presenting them in any text or table. We would be happy to address any specific sources of confusion. 

Table 2: what is the purpose of comparing indigenous versus non-indigenous? This was not highlighted in the introduction.

Response: Aboriginal and Torres Strait Islander Australians experience significant socioeconomic disadvantage, high rates of comorbidity and are more likely to reside in rural and remote areas, where residents may face challenges in accessing healthcare. It is intuitive that therefore that Indigenous Australians may be at greater risk of CNS infection. The reviewer may have missed that this was, in fact, discussed in the introduction in the original submission (lines 88-95 and lines 105-111). 

Species should not be italicised.

Response: The PLOS ONE submission guidelines state that species names should be written in italics. We are very happy for the Editorial staff to advise of their preference.

About enterovirus, was it only one genus identified? What method was used for the identification?

Response: Cases of enterovirus infection were identified using polymerase chain reaction assays. They were not further characterised as this has no impact on clinical management of these patients, which is supportive.

There were too many tables provided, which I think some are not necessary especially the impact of residential address. It would be good if geographic information system is utilized.

Response: We thank the reviewer for their constructive comment. We have moved Table 9 to the supplementary material accordingly (S9 Table). 

We feel that residence in a remote location is actually an interesting variable to examine, as there is significant diversity in the different ecosystems across the 280,000km2 of Far North Queensland. The region includes rainforest, marine environments, lush arable land and dry savannah. It also contains a modern, urban region - Cairns - which is a major international tourist hub. It might be anticipated that residents in different parts of the region would be at risk of CNS from different pathogens due to differences in environmental exposure (eg Burkholderia pseudomallei, Cryptococcus gattii, Leptospirosis, Acanthamoeba), vector exposure (flavivirus, rickettsial disease) or animal exposure (Hendra virus, Australian bat lyssa virus). However, as table 4 demonstrates there are similar rates of most CNS infections in urban and remote settings. 

We believe that this is an important negative finding as national guidelines counsel clinicians to test for many exotic pathogens in people living in remote locations. Our work suggests that these pathogens are, in fact, very rare. Clinicians working in the region should not necessarily waste time and money seeking these pathogens - focussing instead on more common pathogens - in the absence of another clear indication.

We have however adjusted the heading of this paragraph to “Impact of remote residence on disease incidence” to emphasise that it is residence in a remote location, rather than residential address per se (which might suggest an analysis of spatial clustering).

Line 312-313: statistical results should be provided.

Response: The p-values for these statements are presented in Table 8. We have not presented them in the text again as we feel that it would be duplicative, although we would be happy to if the Editor felt that this improved the legibility of the paragraph. 

Discussion: Line 379: Spelling error was observed, N. meningitides

Response: We thank the reviewer for highlighting this oversight. We have revised the paper accordingly (line 403).

Table 9 should be de-tabulated, and the authors should use all the points for the discussion. The table is more appropriate for a review paper.

Response: We thank the reviewer for their constructive comment. We have revised the paper accordingly and moved Table 9 to the supplementary material (S9 Table). 

Epidemiological parameters were not critically discussed in relation to CNS infections. 

Response: We have discussed several epidemiological parameters including age, Aboriginal and Torres Strait Islander status and geographical location throughout the text. It is true that there may be other parameters in the dataset, such as the presence of specific comorbidities that we have not analysed. However, these parameters may not add further value to the paper, and we are concerned that our simple messages may be lost if we try and present all this information in a single manuscript. 

Were there any specific epidemiological parameters that the reviewer thinks it would be useful to look at? 

Lack of references for the comparison was observed.

Response: We are uncertain about what aspect of the manuscript the Reviewer is referring to here. We would be happy to address this point if their specific concern could be made clearer.

---

## [Decision Letter · Decision Letter 1]

23 Feb 2022

PONE-D-21-35660R1The epidemiology and outcomes of central nervous system infections in Far North Queensland, tropical Australia; a 20-year retrospective studyPLOS ONE

Dear Dr. Gora,

Thank you for submitting your manuscript to PLOS ONE. After careful consideration, we feel that it has merit but does not fully meet PLOS ONE’s publication criteria as it currently stands. Therefore, we invite you to submit a revised version of the manuscript that addresses the points raised during the review process.

We look forward to receiving your revised manuscript.

Kind regards,

Latiffah Hassan

Academic Editor

PLOS ONE

Journal Requirements:

Reviewers' comments:

Reviewer's Responses to Questions

**Comments to the Author**

1. If the authors have adequately addressed your comments raised in a previous round of review and you feel that this manuscript is now acceptable for publication, you may indicate that here to bypass the “Comments to the Author” section, enter your conflict of interest statement in the “Confidential to Editor” section, and submit your "Accept" recommendation.

Reviewer #2: All comments have been addressed

Reviewer #3: (No Response)

2. Is the manuscript technically sound, and do the data support the conclusions?

Reviewer #2: Yes

Reviewer #3: Yes

3. Has the statistical analysis been performed appropriately and rigorously? 

Reviewer #2: Yes

Reviewer #3: Yes

4. Have the authors made all data underlying the findings in their manuscript fully available?

Reviewer #2: Yes

Reviewer #3: Yes

5. Is the manuscript presented in an intelligible fashion and written in standard English?

Reviewer #2: Yes

Reviewer #3: Yes

6. Review Comments to the Author

Reviewer #2: The authors have adequately addressed all the previous comments by the reviewer. The revised manuscript has some merits for the reader as the results are well presented and well-concluded. Congratulations!

Reviewer #3: An important retrospective study is presented. The authors have responded well to the comments from the reviewers.

The manuscript is now acceptable for publication. I have only a few minor comments and recommendations:

Recommend change of the end of the title from " a 20 -years retrospective study" to " 2000- 2019"

In the Results section in Abstract: Remove all the quata of 725 from 561/725 etc and present only percentages. The number 725 is presented in the beginning of the results.

Remove decimals on percentages when n<100 :

Rows 268, 271, 280, 290, 292, 293, 308, 319, 321, 336 and on

Table 4 on the group Remote areas (n=80)

Table 8 on Died (n= 32)

7. PLOS authors have the option to publish the peer review history of their article (what does this mean?). If published, this will include your full peer review and any attached files.

Reviewer #2: **Yes: **Rukman Awang Hamat

Reviewer #3: **Yes: **Rune Andersson

---

## [Author Response · Author response to Decision Letter 1]

24 Feb 2022

We thank the editorial staff and reviewers for the time that they have taken to review our manuscript and the helpful comments that they have made to improve the work. Please find below our point-by-point responses to their comments. 

REVIEWERS COMMENTS

Reviewer #2: The authors have adequately addressed all the previous comments by the reviewer. The revised manuscript has some merits for the reader as the results are well presented and well-concluded. Congratulations!

Response: Thank you very much. 

Reviewer #3: An important retrospective study is presented. The authors have responded well to the comments from the reviewers.

The manuscript is now acceptable for publication. I have only a few minor comments and recommendations:

Recommend change of the end of the title from " a 20 -years retrospective study" to " 2000- 2019"

In the Results section in Abstract: Remove all the quata of 725 from 561/725 etc and present only percentages. The number 725 is presented in the beginning of the results.

Remove decimals on percentages when n<100 :

Rows 268, 271, 280, 290, 292, 293, 308, 319, 321, 336 and on

Table 4 on the group Remote areas (n=80)

Table 8 on Died (n= 32)

Response: We thank the reviewer for their comments. This is really a question of style, but we have revised the text as suggested (lines 4, 36-58, 251, 256-258, Table 4, 294, 297, 306, 316, 321, 323, 324, 331, 343, 354, 356, 363, 376, Table 8).

---

## [Editor Report · Decision Letter 2]

2 Mar 2022

The epidemiology and outcomes of central nervous system infections in Far North Queensland, tropical Australia; 2000-2019

PONE-D-21-35660R2

Dear Dr. Gora,

We’re pleased to inform you that your manuscript has been judged scientifically suitable for publication and will be formally accepted for publication once it meets all outstanding technical requirements.

Kind regards,

Latiffah Hassan

Academic Editor

PLOS ONE
---

## [Editor Report · Acceptance letter]

11 Mar 2022

PONE-D-21-35660R2 

The epidemiology and outcomes of central nervous system infections in Far North Queensland, tropical Australia; 2000-2019 

Dear Dr. Gora:

I'm pleased to inform you that your manuscript has been deemed suitable for publication in PLOS ONE. Congratulations! Your manuscript is now with our production department. 

Kind regards, 

on behalf of

Prof Dr. Latiffah Hassan 

Academic Editor

PLOS ONE